# Single Nucleotide Polymorphisms in microRNA Genes and Colorectal Cancer Risk and Prognosis

**DOI:** 10.3390/biomedicines10010156

**Published:** 2022-01-12

**Authors:** Maria Radanova, Mariya Levkova, Galya Mihaylova, Rostislav Manev, Margarita Maneva, Rossen Hadgiev, Nikolay Conev, Ivan Donev

**Affiliations:** 1Department of Biochemistry, Molecular Medicine and Nutrigenomics, Medical University of Varna, 9000 Varna, Bulgaria; galya.mihaylova@mu-varna.bg; 2Laboratory of Molecular Pathology, University Hospital “St. Marina”, 9000 Varna, Bulgaria; 3Department of Medical Genetics, Medical University of Varna, 9000 Varna, Bulgaria; mariya.levkova@mu-varna.bg; 4Department of Oncology, Medical University of Varna, 9000 Varna, Bulgaria; rostislav.manev@mu-varna.bg (R.M.); margarita.bogdanova@mu-varna.bg (M.M.); nikolay.conev@mu-varna.bg (N.C.); 5Clinic of Medical Oncology, University Hospital “St. Marina”, 9000 Varna, Bulgaria; 6Department of Anatomy and Histology, Pathology and Forensic Medicine, Sofia University “St. Kliment Ohridski”, 1000 Sofia, Bulgaria; hadjiev3@gmail.com; 7Clinic of Medical Oncology, Hospital “Nadezhda”, 1000 Sofia, Bulgaria; ivan_donev75@abv.bg

**Keywords:** single nucleotide polymorphism, SNP, miRNA, colorectal cancer

## Abstract

There is growing interest in single nucleotide polymorphisms (SNPs) in the genes of microRNAs (miRNAs), which could be associated with susceptibility to colorectal cancer (CRC) and therefore for prognosis of the disease and/or treatment response. Moreover, these miRNAs-SNPs could serve as new, low-invasive biomarkers for early detection of CRC. In the present article, we performed a thorough review of different SNPs, which were investigated for a correlation with the CRC risk, prognosis, and treatment response. We also analyzed the results from different meta-analyses and the possible reasons for reported contradictory findings, especially when different research groups investigated the same SNP in a gene for a particular miRNA. This illustrates the need for more case-control studies involving participants with different ethnic backgrounds. According to our review, three miRNAs-SNPs—miR-146a rs2910164, miR-27a rs895819 and miR-608 rs4919510—appear as promising prognostic, diagnostic and predictive biomarkers for CRC, respectively.

## 1. Introduction

Colorectal cancer (CRC) ranks as the second leading cause of cancer-related mortality in western countries and the third most common malignancy globally [1]. The majority of CRC cases are “sporadic” when the disease develops with no apparent hereditary syndrome (only 2–8% of cases are, in fact, hereditary) [1]. According to the current oncological standards, CRC prognosis depends on the time of diagnosis as early CRC screening is related to reduced mortality. Colonoscopy, faecal occult blood testing (FOBT) and faecal immunochemical test (FIT) are the most commonly used screening tests worldwide, but each one has specific limits. FOBT and FIT have low sensitivity in pre-neoplastic lesion detection and a high false positive detection rate. Furthermore, colonoscopy is an invasive procedure with poor compliance among those eligible for CRC screening [2,3]. Therefore, there is a need for new reliable biomarkers, which can predict risks of early recurrence and metastasis in CRC patients, biomarkers that could be easily incorporated into the routine diagnostic workup [4]. Considerable attention has been paid to discovering the key role of non-protein-coding RNAs in the regulation of well-known oncogenic and tumor-suppressor signaling pathways at the post-transcriptional level in the context of CRC initiation and progression.

Over the last few years, it has become evident that changes in the expression of several miRNAs (a type of non-coding RNA, 22–25 nucleotides in length) were related to the expression of cancer phenotype via control of fundamental cellular processes such as proliferation, differentiation, migration, angiogenesis, and apoptosis [5]. This is the reason why miRNAs are appealing targets for innovative therapeutic approaches. Additionally, miRNA secretion into extracellular fluids makes them a promising biomarker for evaluating tumor development, progression, and metastasis.

The presence of a single nucleotide polymorphism (SNP) in miRNA genes may explain the deregulated miRNA expression in cancers. Even if the polymorphism does not affect the level of expression of the mature miRNA, it may reflect on the regulatory functions of miRNA on its target genes. The miRNA genes polymorphisms may modify the transcription process on several levels: firstly, on the transcription level of the primary transcript as SNPs in miRNA biogenesis genes and pri-miRNA SNPs; secondly, on the level of pri-miRNA and pre-miRNA processing as SNPs in miRNA biogenesis genes and pri-, pre-miRNA SNPs; thirdly, on the level of miRNA–mRNA interactions as SNPs in mature miRNA sequences and miRNA-binding sites [6,7].

SNPs may be located in different important regions, for example, the promoter regions of the miRNA genes, where they affect the expression of mature miRNAs [8] or in other critical regions as the so-called “seed” sequence, which is essential for the binding of the miRNA to the mRNA (Figure 1). The seed region is a conserved sequence that is mainly situated in positions 2–7 from the miRNA 5’-end [9]. SNPs in this region may affect miRNA complementarity with target genes and lead to deregulation of multiple cellular pathways [10]. SNPs in the miRNA gene region could impact the transcription of miRNAs and the processing to mature miRNAs and, as a consequence, influence miRNA function, stability, and targeting [11]. Moreover, SNPs in miRNA genes or miRNA machinery may affect CRC risk, prognosis, and treatment response [10]. However, Mullany et al. (2016) suggested that SNPs also influence cancer but not through miRNAs, and other mechanisms different to SNPs may control mature miRNA levels [12].

The aim of the review is to summarize and present polymorphisms in miRNAs genes that have potential as prognostic biomarkers and as biomarkers for risk prediction of sporadic CRC.

## 2. Single Nucleotide Polymorphisms (SNPs) in Genes for miRNAs, Associated with Colorectal Cancer (CRC) Risk

The evaluation of the associations of investigated SNPs with CRC risk are presented in Table 1. We found 15 polymorphisms in miRNA genes, for which there are publications about significantly increased CRC cancer risk in their carriers—miR-146a rs2910164, miR-196a-2 rs11614913, miR-608 rs4919510, miR-499 rs3746444, miR-27a rs895819, miR-149 rs2292832, miR-34b/c rs4938723, miR-423 rs6505162, miR-1307 rs7911488, miR-618 rs2682818, miR-492 rs2289030, miR-124-1 rs531564, miR-603 rs11014002 and miRNA-143/145 rs353293.

### 2.1. miR-146a rs2910164

Rs2910164 in pre-miR-146a is the most investigated polymorphism considering the risk of CRC. It results in a change of a G:U pair to a C:U mismatch [13]. The C allele decreases pri-miR-146a nuclear processing efficiency, leading to a less stable secondary structure and reduced expression of mature miR-146 [14]. Santos et al. (2020) demonstrated the functional significance of rs2910164 in CRC patients by finding that miR-146a expression in tumor tissues is significantly higher in patients with GG genotype compared to patients harboring GC or CC genotypes [15]. The link between the high risk of CRC and the presence of the C allele has its explanation. If the C allele causes low levels of miRNA-146a expression, this will lead to less efficient inhibition of its target genes involved in tumorigenesis. However, in the study of Iguchi et al., 2016 CRC cell lines with the pre-miR-146a/C genotype had a significantly higher miR-146a expression than those with the pre-miR-146a/G. The authors tried to explain this different result with ethnicity and disease status [16]. As a functional polymorphism, rs2910164 was studied for a possible association with CRC risk in numerous studies. Nevertheless, most of these studies did not pay attention to the expression level of miR-146a in the investigated cohorts. We compared eight studies that reported that rs2910164 might confer susceptibility to CRC (Table 1). Only two presented an established link between rs2910164 and CRC susceptibility in Caucasian cohorts [17,18]. However, their results were contradictory. Kupcinskas et al. (2014) demonstrated unconvincingly that the C allele was associated with a lower risk of CRC compared to the G allele, while in the study of Dikaiakos et al. (2015), carriers of CC genotype and C allele had a significantly higher risk of CRC. There are also conflicting results of studies among Asian cohorts for the link between rs2910164 and CRC. Chae et al. (2013) suggested a high risk of CRC for carriers of CC genotype compared to those carrying GC or GG genotypes separately and in the recessive model (CC vs. GC + GG) [13]. The C allele also correlated with CRC regardless of whether the tumor was in the colon or the rectum [13]. Another study reported that carriers of the C allele with CC + GC genotypes had an increased likelihood of developing CRC compared to those carrying GG genotype, but this correlation existed only among non-alcohol drinkers [19]. Ma et al. (2013) also investigated the effect of alcohol drinking status on the association between rs2910164 and the risk of CRC [20]. However, they found that non-alcoholic drinkers with CC/CG genotypes had a decreased risk of developing CRC. Moreover, Ma et al. (2013) found that this low risk of CRC in carriers of C allele was more prominent among older individuals, non-smokers, non-drinkers, and no family history of cancer, in intermediately differentiated CRC, and in patients with the advanced stage tumor (Dukes C or D stage) [20]. Asians, harboring the CC genotype and C allele or only male individuals with CC genotype or C allele as well as carriers of GC in GC vs. GG genetic model also had a reduced risk of CRC in studies by Lv et al. (2013), and Hu et al. (2014), respectively [21,22]. Noticeably, the GG genotype was associated with a high risk of CRC in males [23] and females [24] in two different Asian cohorts. According to our knowledge, there are nine other studies that found that rs2910164 was not related to CRC risk (Table 1) [15,24,25,26,27,28,29,30,31].

Rs2910164 is a well-investigated polymorphism, and findings from the individual case-control studies are included in several meta-analyses. Two of them found that G allele and GG genotype in GG vs. GC + CC, [32,33] GG vs. CC [32] and GG vs. GC [33] genetic models were susceptible to CRC. In contrast, stratification analysis by race in another study revealed that GC vs. CC genetic model was associated with a decreased risk of disease in Caucasians [34]. Park et al. (2020) also reported protective effect of CG genotype against digestive system cancers overall in the Europeans, especially for CRC [35]. Liu et al. (2015) summarized eleven studies and demonstrated an association between CC vs. GC + GG genetic model and CRC risk, but only in hospital-based studies [36]. Other meta-analyses did not find an association between rs2910164 polymorphism and CRC risk either in the overall analyses or subgroup analyses based on race [31,37,38,39,40,41,42,43,44].

### 2.2. miR-196a-2 rs11614913

The localization of rs11614913 affects both the maturation of miRNA-196a-2 and its interactions [45]. Several studies link rs11614913 polymorphism with the risk of CRC [21,24,45,46,47,48]. On the other hand, 10 studies do not confirm this association [17,18,25,26,27,29,30,49,50].

Studies that demonstrate an association between the genetic variant rs11614913 and CRC risk could be divided into two groups—studies that identify the C allele as a risk factor and studies that link the T allele to the disease. Zhan et al. (2011) were the first to report that carriers of the C allele with genotypes CT and CC have a higher risk of CRC [45]. They also found that expression levels of miRNA-196a-2 in tumor tissue of patients harboring the C allele were higher than in patients with the TT rs11614913 genotype. The C allele in CT and CC genotypes was also associated with an increased risk of CRC in a study from another Asian cohort [46]. Furthermore, this association was more evident in patients with advanced stages of CRC (Dukes C and D stages) [46]. Min et al., 2012 found that only the CC genotype correlated to an increased risk of CRC in the recessive model (CC vs. CT + TT) and also was statistically significant in females and patients with rectal cancer [24].

The T allele of rs11614913 in heterozygotes (CT vs. CC), homozygotes (TT vs. CC), and in an allelic model in the study by Lv et al. (2013), or T allele in all genetic models in the studies by Toraih et al. (2016) and Haerian et al. (2018) was associated with a high risk of CRC [21,47,48]. Interestingly, in the study by Toraih et al. (2016) the authors reported that the T allele was susceptible to CRC, although the carriers of the C allele had higher expression levels of mature miRNA-196a-2. They explained this contradiction with the specific expression of miRNAs in different cell types. In addition, they claimed that the increased risk of CRC associated with the T allele of rs11614913 was not mediated by differential expression of the mature miRNA-196a-2.

Rs11614913 is the most frequently analyzed SNP in meta-analyses from all presented polymorphisms in this review. Meta-analyses till 2016 show an association between rs11614913 and CRC risk [33,36,37,38,40,41,43,51,52,53,54,55,56,57]. After performing stratified analyses by race, this susceptibility was more prominent for individuals of Asian descent, and none of the genetic models in Caucasians demonstrated an association with the disease risk [37,40,41,42,43,51,54]. Some meta-analyses, mainly after 2016, did not find any relationship between rs11614913 and CRC [32,34,44,58]. The study of Haerian et al. (2018) presented new data from their study on Iranian Caucasians, and in the subsequent meta-analysis, authors found a significant association of rs11614913 and CRC in Caucasians. In contrast, in Asians, and overall, the polymorphism was not relevant to CRC [47]. Moreover, Haerian et al., 2018 showed that the T allele was a risk factor for CRC, which was reported as protective against the disease in other meta-analyses. The last extensive analysis of all studies about the link between rs11614913 and CRC demonstrated that the T allele correlated with an increased risk of CRC under the recessive model (TT vs. CT + CC) [59].

The number of studies for rs11614913 on Caucasians is limited, and there is only one study on Africans. Therefore, future case-control studies on these races could shed light on the conflicting results.

### 2.3. miR-608 rs4919510

Rs4919510 is a cytosine (C) to guanine (G) single nucleotide variation in pre-miR-608. This SNP probably influences miR-608 interactional activities to its target genes. Zhu et al. (2020) found that the G variant allele could increase the expression of the MRPL43 gene, coding for a mitochondrial ribosomal protein [60]. At the same time, the authors did not indicate a direct role of abnormal miR-608 expression.

There is a small number of case-control studies about the association of rs4919510 with CRC risk, but these studies present data on the frequency of this polymorphism among Caucasians (Europeans and Iranians), African Americans, and Asians. [17,28,61,62]. In three of them, there was no significant result, and the correlation between CRC risk and rs4919510 was denied [17,61,62]. Interestingly, Ying et al. (2016) reported that the G allele in the rs4919510 was related to a lower risk of CRC and this finding was more definite for 0-II stages of CRC [28].

There are also several meta-analyses regarding the role of the rs4919510 polymorphism in CRC. Unfortunately, their number is relatively small, which could be explained by the lack of more case-control studies. Nevertheless, two studies found that the G allele in GG vs. CC and GG vs. CC + CG genetic models [63] or only in GG vs. CC genetic models [64] could lead to a lower risk of CRC. On the other hand, according to the other three meta-analyses by Hu et al. (2014), Rong et al. (2017), and Li et al. (2018) there was no significant correlation between the rs4919510 and any specific type of cancer, including CRC [32,65,66].

### 2.4. miR-499 rs3746444

The rs3746444 variant in pre-miR-499 is another example of a SNP, which could be associated with CRC. It involves a change of adenine (A) for guanine (G) [44]. There is a small number of case-control studies regarding the potential role of the rs3746444 polymorphism as a risk factor for CRC. Only Vinci et al. (2013) found out that the prevalence of the homozygotes GG was significantly higher among the CRC patients, and they had a reduced expression of miR-499 [26]. In spite of this positive correlation between the rs3746444 SNP and the risk of CRC, there are studies which deny these findings [18,21,22,24,28,29].

Although only one study reports the association between rs3746444 and CRC risk, and it is among Caucasians, this polymorphism was included in several meta-analyses. Most of them did not find an overall association between the polymorphism and the CRC in all genetic models [33,34,37,38,40,44,52,57,67]. There are two meta-analyses summarizing the studies about the frequency of rs3746444 in CRC patients that find a link between that variant and the disease [36,68]. Liu et al. (2015) firstly found no correlation between rs3746444 and the risk of CRC, but after stratified analysis according to race, they reported a protective effect of the SNP in homozygote Caucasians (GG vs. AA) [36]. Li et al. (2013) concluded that the rs3746444 variant was related to an increased risk of CRC in all homozygote carriers GG [68]. All these contradicting results from the meta-analyses illustrate the need for more case-control studies, especially involving different race groups.

### 2.5. miR-27a rs895819

The rs895819 is a single nucleotide transition of adenine (A) to guanine (G) in a terminal loop of pre-miR-27a that may contribute to abnormal expression and/or secondary structure of miRNA-27a. Rs895819 could reflect on the expression of miRNA-27a’s targeted genes. This polymorphism was the object of several studies investigating its possible correlation with CRC risk. Hezova et al. (2012) were among the first to investigate this SNP’s role in the etiology of colorectal cancer. Their results showed no difference between the genotypes of the cases and the controls and concluded that this polymorphism was not associated with an increased risk of CRC [25]. Kupcinskas et al. (2014) also did not report any correlation between this SNP in pre-miR-27a and an elevated risk of CRC. Nevertheless, the mutated allele was more common among the case group compared to the control one [17]. However, there is one thing in common between these two studies, which deny the role of rs895819 in the etiology of cancer—in both of them, the participants were from Central Europe. Most of the articles, which are in favour of this SNP, recruited patients from Asia. Cao et al. (2014) were one of the first research groups, which presented a correlation between rs895819 and CRC risk. They also measured the expression level of miR-27a in CRC tissue and found that the levels of miR-27a were significantly elevated in carriers of the G allele in both homozygous and heterozygous states, compared to the AA individuals. They proved the functional significance of miR-27a by finding that subjects with genotypes AG and GG were more predisposed to CRC than the AA homozygotes. Also, the risk was higher for males and advanced age (over 60 years old) [69]. Wang et al. (2014), Bian et al. (2015), and Jiang et al. (2016) also investigated the rs895819 polymorphism for association with CRC risk. They all concluded that the GG homozygotes were more predisposed to CRC [70,71,72]. Interestingly, the three studies linked this polymorphism to the progression of the disease. According to Wang et al. (2014), carrying the G allele in one or two copies was associated with a more considerable susceptibility to metastasis [70]. Genotypes AG and GG and allele G correlated to the third stage of CRC in the study by Bian et al. (2015), and GG genotype correlated to the size of the tumor and advanced stage of the disease in the study by Jiang et al. (2016) [71,72]. In spite of that, in one of the latest published research works, the authors did not show an association between the rs895819 and the risk of CRC in the investigated Asian cohort [73]. Zhang et al. (2020) also undertook a pooled analysis, which grouped the results from five different studies, all involving patients from China. They found a significant association between the GG genotype and the risk of CRC, compared to AA individuals and assumed that this SNP might play a role in the etiology of CRC. Zhang et al. (2020) demonstrated the association of rs895819 with CRC in homozygous (GG vs. AA), dominant (GG + AG vs. AA), recessive (GG vs. AG + AA), and allele (G vs. A) inheritance models [73]. In 2016 Feng et al. (2016) and Liu et al. (2016) reported a predisposition of these four genetic models to CRC in their meta-analyses [74,75]. Chen et al. (2017), Zhang et al. (2017), Dai et al. (2019), and Yang et al. (2020) demonstrated that rs895819 could increase the susceptibility to CRC only in homozygotes for the G allele and in the recessive model [76,77,78,79]. Pan et al. (2016) and Alidous et al. (2018), on the other hand, found an association of rs895819 variant with CRC in homozygous state GG allele and under GG + AG vs. AA model as well [34,44]. In an early study by Yuan et al. (2015), only GG homozygotes had a high risk of CRC in the Chinese population [80]. Rs895819 was found to determine CRC risk in Asian carriers also in studies by Liu et al. (2016), Feng et al. (2016), Alidous et al. (2018), and Yang et al. (2020) [34,74,75,79].

No significant association between rs895819 and CRC risk was observed in meta-analyses of Ma et al. (2013) and Rong et al. (2016) [32,33]. Shankaran et al. (2020) also did not report a correlation between the rs895819 polymorphism and predisposition to CRC alone in their case-control study. However, they concluded that this polymorphism together with environmental risk factors and comorbidities might increase the risk of gastrointestinal cancers, including CRC [81].

### 2.6. miR-149 rs2292832

The rs2292832 is a functional polymorphism of miR-149 due to a consecutive nucleotide substitution of thymine (T) with cytosine (C) in chromosome 2. Several research groups have investigated its possible correlation with susceptibility to CRC. Most of them deny the association between the rs2292832 polymorphism and the risk of CRC unequivocally [21,26,29,82]. Vinci et al. (2013) checked the expression of miR-149 in both cancerous and healthy tissue from the index patients [26]. Neither miR-149 expression level nor rs2292832 polymorphisms showed significant associations with colorectal cancer risk in their study. Min et al. (2015) found that rs2292832 polymorphism could modify the risk of CRC in patients younger than 62 years [24]. Ranjbar et al. (2018) reported a significant association between the TT homozygotes and the stratified risk of CRC for sex and age [62]. Chayeb et al. (2018) was the only research group, which found a significant association between the rs2292832 and the risk of CRC—the T allele was more common among the patients with CRC, and the number of the homozygote’s TT was also higher among the case subjects. The authors also reported a significant protective effect of rs2292832 against metastasis when analyzing the over-dominant inheritance model. However, this finding was marginally significant [30]. The significant reported association of the miRNA-149 polymorphism and the risk of CRC in study of Chayeb et al. (2018) could be explained by the ethnic background of the participants studied because the other research groups, which denied the role of miRNA-149 for the susceptibility to CRC, included patients with European or Asian background.

Case-control studies for the role of rs2292832 in CRC are few and mainly denied the association between CRC risk and polymorphism, but meta-analyses that summarized them show different results. Some of them denied the role of rs2292832 polymorphism as a risk factor for CRC not only after analyzing the included case-control studies but also after subgroup analysis was performed [33,37,42,43,52,59,68,74,83]. Other studies reported a significant association in the recessive (TT vs. CC + CT) genetic model and confirmed a high risk of CRC [32,34,36,40]. Analyzing the pool data and these for subgroups, based on different races, Du et al. (2014) found that in addition to the recessive (TT vs. CC + CT) model, the heterozygous (CT vs. TT) genetic model also contributed to increased risk of CRC in the overall cohort and only in Asians [40]. Liu et al. (2015) observed an association with disease in the recessive model after stratified analysis based on the source of control [36]. Interestingly, Pan et al. (2016) concluded that the rs2292832 polymorphism might decrease the risk of CRC both in heterozygotes (CT) and homozygotes (TT) [44].

### 2.7. miR-34b/c rs4938723

Rs4938723 was discovered in the promoter region of pri-miR-34b/c, and the T allele to C allele shift was predicted to influence the GATA-X transcription factors binding sites. Gao et al. (2013) reported that rs4938723 C allele and CC genotype were significantly associated with a decreased risk of CRC, suggesting that this SNP was a protective factor [84]. Kassim et al. (2019) also found the protective effect of CC and TC genotypes of rs4938723. Moreover, the authors demonstrated that the TT genotype, as a risk factor for CRC, was associated with an unhealthy lifestyle and worse clinical features like older age, drinking status, and advanced TNM stage [85]. Oh et al. (2014) showed that rs4938723 alone was not associated with CRC risk, but only if it was combined with TP53 Arg72Pro polymorphism [86]. The authors reported that combinations of rs4938723 TT genotype with TP53 Arg72Pro GC genotype or rs4938723 CC genotype with TP53 Arg72Pro GG genotype were associated with a decreased CRC risk. Additionally, Oh et al. (2014) found that TP53 Arg72Pro CC genotype and dominant model (GC + CC) were significantly related to a reduced CRC risk, which downplays the importance of rs4938723 as a factor for the development of CRC. All three studies did not examine altered miR-34b/c expression levels due to rs4938723 presence.

Three meta-analyses showed a decreased risk of CRC in the homozygote model (CC vs. TT) and in the recessive (CC vs. TC + TT) [87,88,89]. Another meta-analysis revealed that people with the dominant model (TC + TT), homozygous model (TT), and heterozygous model (TC) of genotypes were more susceptible to CRC than carriers of CC genotype [32]. All studies for associations between rs4938723 and CRC were on Asian cohorts. There were no investigations in Caucasians for this SNP hitherto.

### 2.8. miR-423 rs6505162

Jia et al. (2018) reported that rs6505162 AA or AA/CA genotypes had a higher rate of metastasis, but at the same time, the heterozygous genotype showed a significantly decreased risk of CRC [90]. The authors explained this contradiction with the elusive and inconsistent role of miR-423 in cancers in general. A meta-analysis by Moazeni-Roodi et al. (2019) confirmed that rs6505162 significantly decreased the risk of CRC in AC vs. CC and AC + AA vs. CC genetic models [91]. This was also supported by the study of Li et al. (2020), although the authors found that rs6505162 in AA + CA vs. CC and in A vs. C genetic models reduced the risk of digestive cancer as a whole [92]. Two other meta-analyses on the association between rs6505162 and cancer risk found that this polymorphism did not affect CRC susceptibility [77,93].

### 2.9. miR-1307 rs7911488

Rs7911488, located in the terminal-loop of pre-miR-1307, is related to the occurrence and development of CRC. Tang et al. (2015) showed that this SNP caused lower miR-1307 expression as a result of its blockage on the processing done by Dicer1. When the T allele of rs7911488 is replaced by the C allele in pre-miR-1307, MBNL1 (splicing repressor) recognizes it and inhibits Dicer1 recruitment, leading to a low expression of mature miR-1307 [94]. According to the authors, decreased levels of miR-1307 caused the elevation of expression of Bcl2 and the occurrence of CRC. In addition, the C allele of rs7911488 in co-dominant (CC vs. TT), dominant (TC + CC vs. TT), recessive (CC vs. TT + TC) genetic models was a risk factor for CRC. Tang et al. (2015) also found that rs7911488 was markedly associated with tumor diameter. The CC homozygotes in co-dominant and recessive genetic models were more likely to develop larger tumors than TT homozygotes. A new study by Yang et al. (2020) investigated the effect of rs7911488 on CRC growth and metastasis. They reported that T-allelic tumors had a high expression of miR-1307, faster tumor growth, and more metastases than the C-allelic tumors [95]. Interestingly, these contradictory results were published by the same research group. Nevertheless, in the second article, the authors did not explain the significant differences in their results compared to the first one and did not offer any plausible explanation. This illustrates the need for more studies about the potential role of the rs7911488 in the etiology of CRC, especially more case-control studies.

### 2.10. miR-603 rs11014002

Rs11014002 is a scarcely investigated SNP and only in Asian cohorts. It is located within the precursor sequence of miR-603. The C > T exchange enhances the stability of the hairpin structure in a pre-miRNA stem, which probably leads to increased production of the mature miR-603 [96]. Wang et al. (2014) found that CT/TT genotypes were more prevalent among CRC patients than in controls, so the T allele was associated with an increased risk of CRC, especially among non-smokers and non-alcohol drinkers [97]. But Rong et al. (2017) did not find that rs11014002 correlated with the risk of CRC [32].

### 2.11. miR-618 rs2682818

Rs2682818 is located in the hairpin-loop structure of the miR-618 precursor, which may affect the process of miR-618 interaction with its target [98]. Our recent study showed that the deregulated expression of the circulating miR-618 in Caucasian patients with colon cancer was not altered because of rs2682818 variants [99]. We also found that rs2682818 AC genotype was associated with a decreased risk of colon cancer. In concordance to our result, Chen et al. (2018) demonstrated that AA or AC/AA genotype had a lower CRC risk than CC genotype in the codominant model (AA vs. CC) and the dominant model (AC + AA vs. CC) [100].

### 2.12. miR-492 rs2289030

Kupcinscas et al. (2014) were the first to evaluate the association between rs2289030 in the gene for miR-492 and the presence of CRC in the European population, and their study was the only case-control research on this SNP [17]. The authors did not report any significant association between the polymorphism and CRC risk.

### 2.13. miR-124-1 rs531564

Rs531564 (G > C) is a functional SNP in pri-miRNA-124, which affects the expression of the mature miRNA-124 [101]. Qi et al. (2012) found that the presence of the G allele changed the formation of a ring-shaped structure in the secondary structure of the pri-miRNA, and the carriers of the G allele had an increased amount of tumor suppressor miRNA-124 [102]. Rs531564 was significantly associated with protection against colorectal cancer in homozygous codominant (GG vs. CC), dominant (CG + GG vs. CC), recessive (GG vs. CG + CC), and allele (G vs. C) inheritance models in Asians [103]. Moreover, they established a significant association between rs531564 and clinicopathological characteristics of CRC patients, like poor differentiation and lymph node metastasis. At the same time, Ying et al. (2016) did not observe a significant association between the polymorphism and susceptibility to CRC in another large Asian cohort [28]. The authors did not check the association between rs531564 and clinicopathological characteristics, nor whether the rs531564 affected the expression levels of miR-124-1. However, they performed in silico prediction of SNP on miRNA secondary folding and stability, and no significant difference was found in the foldings and their free energies in pre-miR-124-1 carrying wild and mutant allele of rs531564. These different, in terms of their results, studies were included in a meta-analysis, which confirmed that rs531564 G allele was significantly associated with protection against CRC but only under the recessive model (GG vs. CG + CC) [101].

### 2.14. miRNA-143/145 rs353293

Rs353293 G > A in the cluster’s promoter of miR-143/145 induces lower transcriptional activity [104]. Li et al. (2013) established a significant association of the AA genotype or A allele of rs353293 with the increased risk of CRC. Despite that, they did not prove the SNP effect by examining the miR-143/145 expression level [105]. We also found that A allele was a risk factor for CRC in a homozygous state in the investigated Caucasian cohort, and reduced serum levels miRNA-143 and miR-145 were not associated with this genotype’s presence [106].

## 3. SNPs in Genes for miRNAs, Associated with Prognosis and Treatment Response

Several studies described that SNPs in miRNAs genes were associated with patients’ clinicopathological features like tumor size, localization, differentiation, or metastasis. We collected and summarized the data for the SNPs in miRNA genes that were investigated for an association with CRC progression (Table 1). In Table 1, attention is also paid to the studies that did not find a prognostic significance of the separate SNPs. Although fewer in number the studies, which reported prognostic and predictive significance of miRNA’s SNPs in patients with CRC, were intriguing and highly promising.

### 3.1. miR-146a rs2910164

For rs2910164, two studies consider the prognostic significance of this SNP in CRC patients. Santos et al. (2020) established that among CRC patients in an advanced tumor stage, GG genotype carriers had shorter OS than the GC/CC genotype carriers. Regardless, in CRC patients in stage IV, the frequency of GC or CC genotypes was significantly higher than the frequency of GG genotype [15]. In contrast, Chae et al. (2013) found that the CC genotype of rs2910164 was associated with a worse survival outcome compared to the CG or GG genotypes. Moreover, the multivariate analysis showed that the CC genotype of rs2910164 was associated with a worse RFS and DSS than the G allele in the recessive model, adjusted for patient and tumor characteristics [13]. Differences in these studies might be explained by conflicting data on the frequency of the polymorphism in the patients with CRC, as well as the different races of the investigated individuals. To our knowledge, three studies did not find a significant association between rs2910164 and CRC progression [107,108,109].

### 3.2. miR-196a-2 rs11614913

Rs11614913 in the gene for miR-196a-2 was investigated as a prognostic factor for CRC in several studies of Asian CRC patients. In a study by Pao et al. (2018), patients carrying TT and CC genotypes had poorer survival [110]. Jang et al. (2011) found that the heterozygous TC genotype was a significant risk factor under TC vs. TT and TC + CC vs. TT genetic models for unfavorable OS, but not for RFS of rectal cancer patients [108]. In the same study, the authors failed to identify the rs11614913 as a prognostic biomarker in CRC patients like Lee et al. (2010), Zhan et al. (2011), and Chen et al. (2012) in their studies [45,49,107].

### 3.3. miR-608 rs4919510

Ryan et al. (2012) found a significant association between rs4919510 in the pre-miR-608 gene and CRC survival. According to the race, in Caucasians the homozygous variant genotype GG was associated with significantly low OS in CRC patients. The CG genotype also showed a trend towards a poor outcome, although the comparison to the CC referent genotype was not statistically significant [61]. Another study, performed among CRC patients with stage III disease who underwent 5-FU based chemotherapy, showed that the variant-containing genotypes (CG/GG carriers) exhibited increased risks of both recurrence and death. Moreover, these patients had significantly shorter RFS and OS compared to those with the wild-type genotype [111]. Similar results were published by Ranjbar et al. (2018), who found that the CC genotype of rs4919510 contributed to disease progression in the Iranian CRC patients [62]. On the other hand, three studies conducted in Caucasian cohorts of patients showed that rs4919510 was associated with a decreased risk of recurrence (high PFS) only in patients with stage III cancer who received 5-FU-based chemotherapy. In particular, carriers of the variant G allele had a significantly decreased risk of recurrence compared to CC genotype carriers [109,112,113]. In their meta-analysis about the possible rs4929510 effect on cancer prognosis, Dai et al. (2018) found that GG genotype in the homozygous model was associated with poor RFS, but this observation was not confirmed after performing multivariable analysis [63]. Two studies did not find a prognostic significance of rs4919510 among investigated Asian CRC patients [107,110].

### 3.4. miR-26a-1 rs7372209

For rs7372209, it was found that T allele under homozygote comparison (TT vs. CC) and under recessive (TT vs. CT + CC) genetic model was a risk factor for digestive cancer, but not specifically for CRC [92]. Although the association between rs7372209 and CRC risk has not been proven yet, there are investigations of predictive and prognostic significance for the disease. Boni et al. (2011) found a significant association of rs7372209 with tumor response and time to progression (TTP) in metastatic colorectal cancer (mCRC) patients treated with 5-fluorouracil and irinotecan. The authors showed that genotypes CC and CT were associated with longer TTP and ORR (overall response rate) than the TT genotype [114]. Another study observed worse clinical RFS of CRC patients in the II stage undergoing adjuvant chemo-radiotherapy who harbored T allele under dominant (CT + TT vs. CC) and overdominant (CT vs. CC + TT) genetic models [28]. In comparison, we found in a group of mCRC patients that TT genotype was significantly overrepresented in CRC patients with right-sided colon cancer than in patients with left-sided colon cancer and rectal cancer and associated with a longer mean OS than CT and CC genotypes [115].

### 3.5. miR-219-1 rs213210

Lin et al. (2012) found that CT/TT carriers in stage III CRC patients who received 5-FU based chemotherapy had low OS [111]. The same results were confirmed by Pardini et al. (2015). Furthermore, the authors showed that carriers of the T allele who received 5-FU chemotherapy presented with significantly worse survival and an increased risk of relapse [112].

### 3.6. miR-100 rs1834306

Two studies in Caucasians reported opposite results about the predictive and prognostic value of rs1834306 in the pri-miR-100 gene. Both studies included metastatic CRC patients. Boni et al. (2011) found that CC/CT carriers had a significantly high time-to-progression (TTP) [114]. On the other hand, Lampropoulou et al. (2019) reported that CT/TT genotypes of rs1834306 were associated with a significantly reduced TTP and OS. Another interesting fact found by the study team was that carriers of the T allele of the rs1834306 were more likely not to respond to irinotecan-based therapies [116]. Only one study evaluated the prognostic significance of rs1834306 in Asians but did not find any impact of polymorphism on CRC prognosis [107].

### 3.7. miR-423 rs6505162

Xing et al. (2012) reported that rs6505162 CA and AA genotypes in pre-miR-423 were significantly associated with short OS and RFS. A significant interaction between rs6505162 and chemotherapy was not observed, which meant that both patients with and without chemotherapy had an elevated risk of recurrence and/or death [109]. By contrast, no significant associations of survival with rs6505162 were detected in two different Asian cohorts [107,110].

### 3.8. miR-1307 rs7911488

The C allele of rs7911488 was reported to contribute to colon cancer patients’ insensitivity to capecitabine as the response rate of capecitabine chemotherapy was the lowest for patients with CC genotype in comparison with TT and TC genotype carriers [117]. According to the authors, this insensitivity resulted from the low expression of miR-1307-3p and the consequent high expression of TYMS (Thymidylate synthetase). Moreover, in vitro and in vivo experiments on cancer cells with rs7911488 C allele proved the resistance to 5-FU treatment [117].

### 3.9. miR-492 rs2289030

Combined CG and GG genotype of rs2289030 in pre-miRNA-492 were related to significantly worse progression-free survival than that of the patients with the CC genotype, and there was no difference in OS of patients [107]. Their results for rs2289030 as an SNP related to survival were not confirmed in a multivariate analysis. Xing et al. (2012) and Pao et al. (2018) also did not find an effect of rs2289030 on the prognosis of CRC patients in other Asian cohorts [109,110].

### 3.10. miR-124-1 rs531564

Ying et al. (2016) did not find an association between rs531564 and CRC risk, but their study reported that CG/GG genotypes or G allele carriers had worse RFS than CC genotype carriers. This observation was made among stage 0-I, II, and III subgroups and patients receiving adjuvant chemo-radiotherapy. At the same time, CG carriers had a better objective response rate (ORR) to 5-FU based chemotherapy in comparison to patients homozygous for C or G alleles [28].

### 3.11. miRNA-143/145 rs353293

We reported that rs353293 had a prognostic potential for the Bulgarian cohort of CRC patients for the first time. The results showed that metastatic CRC patients with AA genotype had significantly longer mean OS compared to those with CT and CC genotypes [106].

**Table 1 biomedicines-10-00156-t001:** Resume of the studies on the associations of SNPs in miRNA genes with colorectal cancer (CRC) risk, response to the therapy and disease outcome.

SNP ID * Position	miRNA	CRC Cases/ HC	Country	Race	Genotype/Allele Associated with CRC Risk	Found Associations with Clinicopathologic Features of Patients and CRC Outcome	References
rs2910164 G > C chr5:160485411	miR-146a	197/212	Czech Republic	Caucasians	*no association*	-	[25]
		160/178	Italy	Caucasians	*no association*	-	[26]
		193/428	Lithuania	Caucasians	C—lower risk compared to G	-	[17]
		157/299	Greece	Caucasians	CC/C—high risk	-	[18]
		152/161	Tunisia	Caucasians	*no association*	G/C are associated with tumor differentiation	[30]
		125/276	Brazilia	Caucasians, non-Caucasians	*no association*	GC/CC are associated with advanced CRC and/or metastasis GG carriers in stage IV—short OS GG is associated with a high level of mature miR-146a	[15]
		426/-	Korea	Asians	-	no prognostic significance	[107]
		407/-	Korea	Asians	-	no prognostic significance	[108]
		446/502	Korea	Asians	GG—high risk in female	GG is associated with an increased risk of proximal colon cancer	[24]
		408/-	China	Asian	-	no prognostic significance	[109]
		1147/1203	China	Asians	GC/CC—low risk	GC/CC—histological differentiation and advanced stage	[20]
		353/540	China	Asians	CC/C—low risk	-	[21]
		399/568	Korea	Asians	CC—high risk	CC carriers—short OS, RFS, and DSS	[13]
		554/566	China	Asians	CC/CG carrier, non-alcohol drinkers—high risk	-	[19]
		524/116	Japan	Asians	*no association*	-	[27]
		276/373	China	Asians	GC—low risk	GG/G—better histological differentiation	[22]
		59/-	Japan	Asians	-	GC/CC are associated with tumor progression and synchronous liver metastasis	[16]
		1358/1079	China	Asians	*no association*	-	[28]
		560/780	China	Asians	GG/G—high risk in male	-	[23]
		1003/1303	China	Asians	*no association*	-	[31]
		899/204	Australia, USA, Canada	not available	*no association*	-	[29]
rs11614913 C > T chr12:53991815	miR-196a-2	197/212	Czech Republic	Caucasians	*no association*	-	[25]
		160/178	Italy	Caucasians	*no association*	-	[26]
		193/428	Lithuania	Caucasians	*no association*	-	[17]
		157/299	Greece	Caucasians	*no association*	-	[18]
		152/161	Tunisia	Caucasians	*no association*	-	[30]
		907/1243	Iran	Caucasian	TT—high risk	-	[47]
		194/286	Iran	Caucasian	*no association*	-	[50]
		30/100	Egypt	Africans	CT/TT/T—high risk	C allele is associated with a high level of miR-196a-2	[48]
		426/-	Korea	Asians	-	no prognostic significance	[107]
		252/543	China	Asians	CC/CT—high risk	CC/CT are associated with a high level of miR-196a-2 no prognostic significance	[45]
		407/-	Korea	Asians	-	TC genotype—short OS	[108]
		446/502	Korea	Asians	CC—high risk	CC is associated with rectal tumor localization	[24]
		573/558	China	Asians	CC/CT—high risk	CC/CT are associated with advanced stages	[46]
		126/407	China	Asians	*no association*	no prognostic significance	[49]
		353/540	China	Asians	CT/TT/T—high risk	-	[21]
		524/116	Japan	Asians	*no association*	-	[27]
		188/-	Taiwan	Asians	-	TC/CC—short OS	[110]
		899/204	Australia, USA, Canada	not available	*no association*	-	[29]
rs4919510 C > G chr10:100975021	miR-608	245/446	USA	Caucasians, African Americans	*no association*	GG carriers, Caucasian—short OS	[61]
		1097/-	USA	Caucasians, African Americans	-	CG/GG carriers, stage III patients after 5-FU CT—short OS and RFS	[111]
		193/428	Lithuania	Caucasians	*no association*	-	[17]
		1083/-	Czech Republic	Caucasians	-	CG/GG carriers, stage III patients after 5-FU CT—high PFS	[112]
		76/70	Iran	Caucasians	*no association*	GG—metastatic clinicopathological features	[62]
		426/-	Korea	Asians	-	no prognostic significance	[107]
		408/-	China	Asians	-	CG/GG carriers after *CT*—high PFS	[109]
		1358/1079	China	Asians	GG/G carriers, stage 0-II—low risk	-	[28]
		155/-	UK	not available	-	CG/GG carriers, rectal cancer patients after CT or CT + radiotherapy—high OS and PFS	[113]
		188/-	Taiwan	Asians	-	no prognostic significance	[110]
rs3746444 A > G chr20:34990448	miR-499	160/178	Italy	Caucasians	GG—high risk	G allele is associated with a low level of miRNA-499	[26]
		157/299	Greece	Caucasians	*no association*	-	[18]
		899/204	Australia, USA, Canada	not available	*no association*	-	[29]
		446/502	Korea	Asians	*no association*	-	[24]
		407/-	Korea	Asians	-	no prognostic significance	[108]
		408/-	China	Asians	-	no prognostic significance	[109]
		353/540	China	Asians	*no association*	-	[21]
		276/373	China	Asians	*no association*	-	[22]
		1358/1079	China	Asians	*no association*	-	[28]
rs895819 T > C chr19:13836478	miR-27a	212/197	Czech Republic	Caucasians	*no association*	-	[25]
		193/428	Lithuania	Caucasians	*no association*	-	[17]
		408/-	China	Asians	-	no prognostic significance	[109]
		205/455	China	Asians	GG—high risk	GG/G are associated with an increased risk of metastasis	[70]
		254/238	China	Asians	AG/GG—high risk	AG/GG are associated with older age (≥ 60 years) and male gender GG/AG carriers have high levels of miR-27a	[69]
		412/412	China	Asians	GG—high risk	AG/GG and G are associated with stage III	[71]
		508/562	China	Asians	GG—high risk	GG is associated with a larger tumor size (>5cm), G is associated with a higher pathological stage (TNM-III)	[72]
		208/312	China	Asians	*no association*	-	[73]
		75/204	India	not available	*no association*	-	[81]
rs2292832 T > C chr2:240456086	miR-149	160/178	Italy	Caucasians	*no association*	-	[26]
		76/70	Iran	Caucasians	*no association*	TT is associated with gender and age in patients	[62]
		152/161	Tunisia	Caucasians	TT—high risk		[30]
		446/502	Korea	Asians	*no association*	TT is associated with ages < 62 years of patients	[24]
		407/-	Korea	Asians	-	no prognostic significance	[108]
		443/434	China	Asians	*no association*	-	[82]
		408/-	China	Asians	-	no prognostic significance	[109]
		353/540	China	Asians	*no association*	-	[21]
		899/204	Australia, USA, Canada	not available	*no association*	-	[29]
rs4938723 T > C chr11:111511840	miR-34b/c	347/488	China	Asians	CC/C—low risk	-	[84]
		545/428	South Korea	Asians	TT and CC genotypes in combination with GC and GG genotypes of TP53 Arg72Pro—low risk	-	[86]
		1078/1175	China	Asians	CC/TC—low risk	CC/CT carriers—high MST	[85]
rs7372209 T > C chr3:37969217	miR-26a-1	61/-	Spain	Caucasians	-	CC/CT carriers after 5-FU and CPT-11—high ORR and TTP	[114]
		101/90	Bulgaria	Caucasians	*no association*	TT is associated with right-sided colon cancer and longer OS	[115]
		426/-	Korea	Asians	-	no prognostic significance	[107]
		1358/1079	China	Asian	*no association*	CC carriers stage II patients after CT—high RFS	[28]
rs213210 A > G chr6:33208047	miR-219-1	1097/-	USA	Caucasians, African Americans	-	CT/TT carriers, stage III patients after 5-FU CT—short OS	[111]
		1083/-	Czech Republic	Caucasians	-	CT/TT carriers after 5-FU CT—short OS and PFS	[112]
		426/-	Korea	Asians	-	no prognostic significance	[107]
rs1834306 T > C chr11:122152479	miR-100	61/-	Spain	Caucasians	-	CC/CT carriers—high TTP	[114]
		105/-	Greece	Caucasians	-	CT/TT carriers—short TTP and OS	[116]
		426/-	Korea	Asians	-	no prognostic significance	[107]
rs6505162 A > C chr17:30117165	miR-423	426/-	Korea	Asians	-	no prognostic significance	[107]
		408/-	China	Asian	-	CA/AA carriers—short OS and RFS	[109]
		117/84	China	Asian	AC/AA—low risk	AA and AA/AC- high risk of metastasis	[90]
		188/-	Taiwan	Asians	-	no prognostic significance	[110]
rs7911488 T > C chr10:103394332	miR-1307	1026/1026	China	Asian	C—high risk	CC is associated with larger tumors, C allele is associated with a low level of miR-1307	[94]
		274/-	China	Asians	-	TC—higher than TT RR of capecitabine-based treatment; CC-less sensitive to capecitabine chemotherapy	[117]
rs11014002 C > T chr10:24275724	miR-603	102/204	China	Asian	CT/TT—high risk	-	[97]
rs2682818 A > C chr12:80935757	miR-618	104/90	Bulgaria	Caucasians	AC—low risk	no prognostic significance	[99]
		878/884	China	Asians	AA and AC/AA—low risk	-	[100]
rs2289030 G > C chr12:94834510	miR-492	193/428	Lithuania	Caucasians	C—higher risk compared to G	-	[17]
		426/-	Korea	Asians	-	CG/GG carriers—short PFS	[107]
		408/-	China	Asians	-	no prognostic significance	[109]
		188/-	Taiwan	Asians	-	no prognostic significance	[110]
rs531564 G > C chr8:9903189	miR-124-1	426/-	Korea	Asians	-	no prognostic significance	[107]
		900/1110	China	Asians	CG/GG—low risk	CG/GG carriers—low risk of poor differentiation and lymph node metastasis	[103]
		1358/1079	China	Asian	*no association*	CG/GG carriers stage 0-III patients, after chemo-radiotherapy—short RFS	[28]
rs353293 G > A chr5:149427663	miR-143/145	99/89	Bulgaria	Caucasians	AA—high risk	AA carriers—longer OS	[106]
		242/283	China	Asians	A– high risk	-	[105]

* GRCh38.p13. HC—healthy controls; OS—overall survival; RFS—recurrence-free survival; PFS—progression-free survival; DSS—disease specific survival; MST—median survival time; ORR –overall response rate; TTP—time to progression; CPT-11—irinotecan; RR—response rate.

## 4. Discussion

In our study, we performed a detailed review of SNPs in genes for miRNAs investigated for association with risk of CRC and a relationship with prognosis of the disease and/or treatment response. Interest in studying miRNAs-SNPs has been part of the quest in recent years to find low-invasive, promising new biomarkers for early detection of CRC. Our detailed review highlights the role of three miRNAs-SNPs as potential prognostic, diagnostic and predictive biomarkers. The accumulated data from the studies of miR-146a rs2910164, miR-608 rs4919510, and miR-27a rs895819 in CRC patients from different races and ethnicities showed that GC and CC genotypes of rs2690164 were associated with tumor progression and poor outcomes of CRC patients; Caucasian and Asian patients, carriers of CG or GG rs4919510 genotypes after chemotherapy had high PFS; and G allele of rs895819 was associated with a high CRC risk, increased risk of metastasis, and higher TNM stage at least for Asians.

These SNPs were also investigated for their role in other cancers. Rs2910164 was not only the most studied miRNAs-SNP in CRC but it was also well investigated in other different cancers. The meta-analyses from the last few years demonstrated the association of rs2910164 with the increased risk of lung cancer [118,119], head and neck carcinoma [120], urological neoplasms, particularly bladder cancer [121], gastric cancer [35] and brain tumors [122]. For the last two cancers this risk was linked to GG genotype. However, patients with CC or CG genotypes of rs2910164 were more susceptible to the other cancers. At the same time, three large meta-analyses found that the CC genotype was associated with a decreased risk of cervical, hepatocellular, and prostate cancer [123,124,125].

For rs4919510 Ding et al. (2018) found that had a protective effect on cancer OS under CG vs. GG, CG + CC vs. GG, and CC vs. CG + GG genetic modes and carriers of CG were associated with better RFS in comparison with GG carriers [126]. On the other hand, the same SNP acted as a protective factor against head and neck cancer development under the genetic models—G vs. C and GG + GC vs. CC [127]. At the same time, the variant G allele of rs4919510 exhibited an increased risk of lung and papillary thyroid cancers in heterozygotes [64]. Rs895819 has already been associated with an increased cancer risk among Asians [76,80]. Furthermore, the analyses in two studies showed that harboring of the G allele of rs895819 determined a decreased cancer risk in Caucasians (AG vs. AA; GG + AG vs. AA) [77,128]. Interestingly, the G allele of rs895819 was significantly associated also with an increased risk of lung cancer, but with a decreased risk of breast cancer [76,77,129,130,131]. Rs895819 has been investigated more in patients with breast cancer, however, the results for it were inconsistent. Nevertheless, three meta-analyses including the last from 2021 which claimed to provide a more accurate evaluation of all published eligible studies showed that the wild-type AA genotype rs895819 was associated with increased susceptibility to breast cancer among Caucasians but not among Asians while harboring G-allele and AG genotype might have a protective role [129,130,131]. The C (G) allele of rs895819 was associated with shorter overall survival and increased risk of death than the T (A) allele in Asian patients with non-small cell lung cancer [132,133]. According to Xia et al. (2017), the polymorphism in miRNA-27a could predict not only the outcome of non-small cell lung cancer patients but also might decrease the sensitivity to anti-cancer drugs [132].

There have been more studies on the impact of SNPs in miRNAs genes of the susceptibility to CRC than those that have investigated them as predictors of clinical outcomes in CRC patients. Despite the small number of such studies, several SNPs could be associated with CRC progression. These are miRNA-146a rs2010164, miR-196a-2 rs11614913, miRNA-27a rs895819, and miRNA-423 rs6505162. For all these, there were data that showed their associations with poor outcomes and/or increased risk of metastasis in CRC patients.

Many meta-analyses have summarized these case-control studies regularly, but they often present conflicting reports that do not provide a conclusive result. There could be several possible reasons for this. Firstly, meta-analyses often include studies with high heterogeneity, with stratification biases of their cohorts. Some of the meta-analyses exclude separate publications that lead to other computational or statistical biases accumulating or reducing the effects of individual SNPs in the data analysis. Secondly, SNPs in miRNAs genes probably might lead to an increased incidence of cancer in a completely different manner in divergent races and ethnicities. Most of the studies on this matter include Asian cohorts. Therefore more investigations of the role of miRNAs-SNPs in Caucasian cohorts are needed to define if some genetic variants are specific only for the particular race/ethnicity or they are, in general, common for all. Thirdly, a small number of case-control studies included in meta-analyses trace the impact of interactions between genes and the environment and lifestyle choices of patients. In recent years, several epidemiological studies have shown that in the pathophysiology of many diseases, including cancers, the interactions of environmental factors with specific allelic variants noticeably modulate the susceptibility to diseases.

Many of the research studies looking for a link between CRC and miRNAs-SNPs are descriptive studies without functional experiments. Some of them did not evaluate even the levels of current miRNA, which makes it difficult to assess the functional effect of SNPs. It is possible that a polymorphism has unknown mRNA targets, other than known ones, which could activate or inhibit different tumorigenesis pathways in the cell. We did not find the studies in CRC patients, which explore the role of genetic variants in other miRNAs or genes in similar linkage disequilibria with investigated SNPs. Similar data could shed light on the underlying mechanism of how one miRNA-SNP without functional consequences affects CRC risk.

It is hard to tell if the analysis of miRNAs-SNPs, which is low-invasive, because it requires a one-time blood sample and is relatively easy to perform, could be included in routine practice and how could it be combined with other CRC screening tests. It is probable that studies that encompass and utilize improved statistical tools and computational models which include increased sample sizes would present more powerful and dependable results.

## 5. Conclusions

In the present study, we considered single nucleotide polymorphisms in miRNAs genes for their association with the susceptibility to sporadic CRC and/or for association with disease prognosis and response to therapy. As a result, we identified three SNPs with a proven prognostic, diagnostic and predictive significance, respectively, for the disease. Further experiments are needed to confirm our conclusions for miR-146a rs2910164, miR-608 rs4919510, and miR-27a rs895819 due to a large number of contradictory studies in the literature.

## Figures and Tables

**Figure 1 biomedicines-10-00156-f001:**
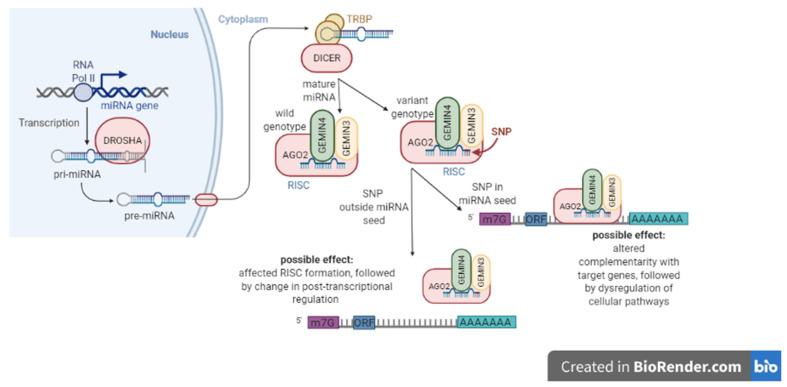
Sequence variations in mature miRNAs. Single nucleotide polymorphisms (SNPs) in miRNA “seed” sequence may lead to a change in miRNA complementarity. Consequently, this may dysregulate multiple cellular pathways. On the other hand, SNPs outside the “seed” region may increase or decrease the efficiency in miRNA binding to mRNA.

## Data Availability

Not applicable.

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
