# Peer review of "Single Nucleotide Polymorphisms in microRNA Genes and Colorectal Cancer Risk and Prognosis"

_biomedicines, 2022, doi:10.3390/biomedicines10010156_

Round 1
Reviewer 1 Report
In this review, the authors have summarized the significance of SNPs in miRNA in the prognosis and treatment outcome of colorectal cancer patients. The manuscript is well written. The following points can improve the manuscript.
- The significance of SNPs in miRNAs should be emphasized in the context of colorectal cancer progression.
- A brief information on the role of SNPs in miRNAs in other cancer types should be provided.
Author Response
Reviewer 1
In this review, the authors have summarized the significance of SNPs in miRNA in the prognosis and treatment outcome of colorectal cancer patients. The manuscript is well written. The following points can improve the manuscript.
Comment1:
1) The significance of SNPs in miRNAs should be emphasized in the context of colorectal cancer progression.
Author’s Reply:
Thank you for your comment. As we mentioned in section "4. Conclusions", the studies on the impact of SNPs in miRNAs genes of the susceptibility to CRC were more than those that investigated them in connection with the progression of the disease.
We have made several additions to the text, emphasizing the importance of miRNAs SNPs in CRC progression.
- From line 431 to line 433 in section “3. SNPs in genes for miRNAs, associated with prognosis and treatment response”;
- From line 500 to line 503 in section “4. Discussion”.
All incorporated changes are presented using the ‘tracked changes’ function in the revised manuscript.
Comment 2:
2) A brief information on the role of SNPs in miRNAs in other cancer types should be provided.
Author’s Reply:
Thank you for this suggestion. We agree that information on the role of SNPs in miRNAs in other cancer types is important to add to the manuscript. However, following the role of all presented in our study SNPs in other cancers would shift the focus of the manuscript. Therefore, we wrote two paragraphs in the new section "4. Discussion" (from line 557 to line 597) with respect only to the role of rs2910164, rs4919510 and rs895819 in other cancers. These are SNPs that we think may have significance for colorectal cancer. All incorporated changes are presented using the ‘tracked changes’ function in the revised manuscript.
We are very thankful for your comments. We are hoping that we have understood them and our answers and revision of the manuscript are acceptable.

Reviewer 2 Report
In this manuscript, the authors performed a detailed review of different miRNA-SNPs, which are investigated for a correlation with the CRC risk, prognosis and treatment response, and highlights that two miRNAs-SNPs: miR-27a rs895819 and miR-608 rs4919510 can be used as biomarkers for diagnosis and prognosis of CRC.
The manuscript is well written and the illustration is presented in a good quality. This will provide interesting information for the reader of the journal. However, there are still some grammatical and syntax errors in the article. So I think the manuscript can be accepted after grammar and language check.
Author Response
Reviewer 2
In this manuscript, the authors performed a detailed review of different miRNA-SNPs, which are investigated for a correlation with the CRC risk, prognosis and treatment response, and highlights that two miRNAs-SNPs: miR-27a rs895819 and miR-608 rs4919510 can be used as biomarkers for diagnosis and prognosis of CRC.
The manuscript is well written and the illustration is presented in a good quality. This will provide interesting information for the reader of the journal. However, there are still some grammatical and syntax errors in the article. So I think the manuscript can be accepted after grammar and language check.
Author’s Reply:
Thank you for your comment. We improved the quality of the English language as much as we could and we believe that we now meet the high standards of the journal.
We are very thankful for your observations and comments.
